# Do Organism Profile and Resistance Patterns Change between First and Subsequent Two-Stage Revision for Periprosthetic Joint Infection?

**DOI:** 10.3390/antibiotics13080771

**Published:** 2024-08-15

**Authors:** Helmut Ahrens, Amelie Constanze Steinicke, Georg Gosheger, Jan Schwarze, Sebastian Bockholt, Burkhard Moellenbeck, Christoph Theil

**Affiliations:** Department of Orthopedics and Tumor Orthopedics, Muenster University Hospital, Albert-Schweitzer-Campus 1, 48149 Muenster, Germany; helmut.ahrens@ukmuenster.de (H.A.); georg.gosheger@ukmuenster.de (G.G.);

**Keywords:** periprosthetic joint infection, prosthetic joint infection, PJI, revision arthroplasty

## Abstract

Increasing antibiotic resistance has been reported as an issue in the treatment of periprosthetic joint infection (PJI). A repeat two-stage revision for recurrent PJI is at high risk of reinfection. However, it is unclear if the microorganism profile plays a role with potentially more resistant or polymicrobial infections. This is a retrospective, single-center analysis of two-stage revisions performed between 2011 and 2017. We identified 46 patients who underwent a repeat resection arthroplasty for recurrent PJI of the same joint after a previous two-stage revision of the same joint at the same department. All microbiological findings were analyzed focusing on microbiological spectrum and resistance testing as well as the potential impact on reinfection-free survival. The most common organism found at the time of recurrent PJI were coagulase-negative *Staphylococci* (39%) followed by Gram-negative organisms (28%). The risk of polymicrobial infections, difficult-to-treat resistant organisms, and Gram-negative infections increased significantly. Among *staphylococcal* infections, there was a high percentage of methicillin-resistant species and resistance to oral antibiotics. Patients with Gram-negative organisms had a reduced infection-free survivorship, while resistant organisms were not associated with decreased survival. Patients who undergo a repeat two-stage revision for recurrent PJI have more polymicrobial and resistant organisms, although the impact on survivorship is unclear.

## 1. Introduction

Periprosthetic joint infection (PJI) is a devastating complication of joint arthroplasty and occurs in around 1–2% of primary knee (TKA) or hip arthroplasties (THA) [1,2,3]. As the demand for total joint arthroplasty is on the rise due to an aging population, the revision burden due to PJI is expected to increase as well [4]. Furthermore, patients who undergo revision arthroplasty are at increased risk of PJI, particularly in patients who had undergone previous surgery for PJI [4,5,6,7,8,9]. In these complex cases, the risk for a recurrent PJI can be greater than 20% [5].

A common approach for the surgical management of chronic infections is a two-stage exchange usually using an antibiotic-loaded spacer [6,10,11,12]. This approach comprises the removal of the implant and all foreign material, debridement, and irrigation during the first stage and subsequent second-stage reimplantation after the completion of systemic antibiotic therapy based on the organisms cultured. However, the rate of reinfection can exceed 30% in some cases. The recurrence of infection is associated with further morbidity as well as a high mortality depending on the management chosen, particularly if further surgeries are performed [6,7,10]. In these cases, one option is to perform a repeat two-stage procedure, particularly if the infection is chronic and desired to ultimately retain a functioning arthroplasty. However, salvage treatment, amputation, and antibiotic suppression are other valid options in this catastrophic scenario [13]. While existing studies have been limited in terms of the size of the studied cohorts and relative scarcity of such patients at individual centers, it is generally agreed that the risk of reinfection is very high with failure rates ranging from 22% to 49% [6,13,14].

One potential reason for failure that has not been extensively studied in the literature is the potential shift in the microbiological spectrum encountered at recurrent PJI as well as the risk of increasing antibiotic resistance due to the repeat exposure to long-term systemic and local antibiotic treatment when performing repeat stage revision arthroplasty [15,16].

One study [15] investigated the potential risk of resistance between the first and second stage of a single two-stage exchange and noted that 7% of 142 patients who underwent a two-stage revision of a hip or knee prosthesis developed relevant increasing antibiotic resistance in between stages. However, it is unclear how antibiotic resistance and the organism spectrum change between two staged revisions of chronic PJIs of the same joint. This study therefore investigates the microbiological spectrum, antibiotic resistance pattern, risk factors for antibiotic resistance, and the potential impact on infection-free survivorship.

## 2. Results

Overall, in comparing the spectrum of organisms cultured, there was a greater variety of organisms at the time of the repeat two-stage exchange (55 vs. 65 different species) (Table 1).

The risk of polymicrobial infections increases at the time of the second two-stage revision (20 vs. 33%, *p* = 0.263).

Furthermore, the likelihood of an infection with a difficult-to-treat organism increased at the time of the second two-stage revision (9 vs. 28%, *p* = 0.049).

The most common organism was coagulase-negative *Staphylococci* for both the first and second two-stage revision (48% and 39%). While *Staphylococcus aureus* was the second most common organism at the first two-stage revision (22%), Gram-negative organisms (*Escherichia coli*, *Proteus*, *Pseudomonas Pasteurella*, *Citrobacter*, *Klebsiella* and *Enterobacter*) were the second most common type of organism at the second two-stage revision (28%). While generally, there were no significant difference for individual organisms, Gram-negative organisms were more common at the second two-stage revision (11 vs. 28%, *p* = 0.008).

For patients who had a PJI caused by coagulase-negative *Staphylococcus*, the rate of resistance to methicillin (MR CoNS) was not different at the time of the repeat two-stage revision (77% vs. 77%, *p* = 0.913) although it was very high. Additionally, 86% of these organisms were resistant to oral antibiotics with the exemption of linezolid.

On the other hand, patients with *Staphylococcus aureus* infections had a higher risk of methicillin resistance at the first two-stage revision (MRSA) (20% vs. 0%, *p* = 0.243).

There were new/different organisms in 78% of patients between the first and second two-stage revision, while 22% of patients had at least one organism identified at both time points (polymicrobial infections and coagulase-negative *Staphylococcus).*

With the numbers available, we were unable to identify potential risk factors for the development of a difficult-to-treat organism. Specifically, there were no differences between men and women (*p* = 0.056), in hip or knee PJI (*p* = 0.738), in obese patients (*p* = 0.181), in diabetics (*p* = 0.505), in age (*p* = 0.142), or in comorbidity score (*p* = 0.09). Furthermore, there was no difference in the number of previous surgeries (*p* = 0.786) or in the reinfection-free period (*p* = 0.786) for patients who developed difficult-to-treat organisms.

Considering that coagulase-negative *Staphylococci* were the most common organism cultured, risk factors for methicillin resistance were analyzed separately. With the numbers available, there were no differences between men and women (*p* = 0.538), in hip or knee PJI (*p* = 1), in obese patients (*p* = 0.117), in diabetics (*p* = 0.761), in age (*p* = 0.573), or in comorbidity score (*p* = 0.632). Furthermore, there was no difference in the number of previous surgeries (*p* = 0.308) or in the reinfection-free period (*p* = 0.878) for patients who developed difficult-to-treat organisms.

The reinfection-free survival probability after the second two-stage exchange amounted to 80% (95% CI 68–92%) after one year and 68% (95% CI 54–82%) after two years.

Patients who had methicillin-resistant coagulase-negative *Staphylococci* had a worse reinfection-free survival probability compared to other infections after two years (51% (95% CI 23–74%) vs. 74% (95% CI 58–90%), *p* = 0.246).

Patients who had a Gram-negative organism cultured displayed a reduced reinfection-free survival probability after two years (42% (95% CI 14–70%) vs. 78% (95% CI 63–92%), *p* = 0.091).

Patients who had a difficult-to-treat organism cultured had a similar reinfection-free survivorship compared to other organisms after two years (64% (95% CI 38–92%) vs. 65% (95% CI 57–82%), *p* = 0.923).

Furthermore, patients with polymicrobial infections had a similar reinfection-free survivorship compared to monomicrobial infections after two years (65% (95% CI 40–88%) vs. 69 (95% CI 52–86%), *p* = 0.997).

## 3. Discussion

This study analyzed the microbiological profile and resistance testing of PJI in patients who underwent a second two-stage exchange for recurrent PJI. This study’s main findings are (1) the risk of polymicrobial infections and organisms that are considered difficult to treat because their resistance to certain antibiotics is increased, (2) a stark increase in Gram-negative infections at the time of the second two-stage exchange and these patients being at a high risk of reinfection, (3) methicillin resistance in staphylococcal infection not changing, and (4) risk factors for increasing resistance remaining unclear.

While this study first investigated antibiotic susceptibility and the microbiology profile in patients with repeat two-stage exchange for recurrent chronic PJI, it has several limitations: firstly, it is limited by small patient numbers that lead to heterogeneity when investigating individual organisms or resistance patterns. Therefore, there may be factors relevant to treatment success that cannot be detected with the study design. Future multi-institutional studies should be performed to include greater numbers of these difficult cases. Furthermore, this study is limited by its retrospective design, relying not only on the completeness of patient records with their inherent weakness but also on patients returning for follow-up visits. While we decided to use a minimum follow-up period of 12 months, which has been considered adequate by previous authors in the setting of PJI [17], it is nonetheless possible that patients who are often comorbid or referred from other institutions might not return to the original provider, particularly if complications occur. Therefore, the reported probability of survival must be considered a low-end estimate, as treatments might have been performed elsewhere.

The role of microbiology findings in patients with recurrent PJI who were treated with repeat revision surgery or repeat two-stage exchange were studied by other groups in the past. One study included 42 patients with a repeat two-stage exchange of an infected hip arthroplasty [14], and it found that only 57% of patients achieved infection control. The authors noted that 50% of patients had pathogen persistence, with coagulase-negative *Staphylococci* being most common. On the other hand, Gram-negative infections were rare and only accounted for 10% of recurrent PJI in their study. The authors commented that the patients included were severely comorbid and had multiple failed attempts of endoprosthetic reconstruction, and it is unclear what role microbiology really plays in these complex scenarios. Fehring et al. [6] investigated 45 patients who had a second two-stage exchange for recurrent PJI of the knee. They found that 49% failed to clear the infection and had a repeat revision performed. They noted a relatively high percentage of culture-negative infections at the time of the second PJI (24%), with coagulas-negative *Staphylococci* being the most common. While microbiology results were not associated with treatment success or failure in their study, it is noteworthy that 44% of patients were on chronic suppression therapy in their study at final follow-up, which was not carried out in this study.

One other study that included 44 patients who underwent repeat revision after an initial two-stage exchange for hip or knee PJI similarly noted a high risk of failure with only 52% of patients undergoing endoprosthetic reconstruction at last follow-up [18]. Contrary to the findings of Khan et al. [14] and the present study, they noted culture negativity in 60% of cases and pathogen persistence in 29%. They hypothesized that early broad-spectrum antibiotic treatment might have been started early if recurrent infection had been suspected, leading to a stark decrease in pathogen detection. However, considering that their study spanned 15 years, starting in 2000, it is possible that microbiology methods have evolved as well. We excluded four patients with culture-negative infections, as the main focus of this study was to investigate resistance and organism details; however, the factors contributing to pathogen persistence remain elusive, and further studies including molecular methods might contribute to a more profound understanding, and to the differentiation between persisting and new infection [19].

Considering that the existing literature [6,14] has found that coagulas-negative *Staphylococci* were most common at the time of recurrent chronic PJI, it is noteworthy that reported results of PJI with this type of organism may be associated with poor outcomes regardless of first or subsequent infection [20,21]. One study investigated 55 patients with a culture-proven CoNS PJI of the knee joint and found that only 47% of patients had reached infection control. Notably, they performed staged exchanges in 55% of their patients, which were associated with an even worse rate of infection control of only 47% with around 2.5 years of follow-up. And they noted that 63% of organisms cultured were methicillin-resistant CoNS. While antibiotic susceptibilty and choice were not associated with infection control in their study, they noted that prolonged oral antibiotic treamtent was an important determinant for success. Contrary to these findings, one study [21] on 111 patients with chronic PJI with CoNS or other resistant organisms treated orally with an antibiotic regimen containing linezolid and a two-stage exchange found a long-term infection control of 77% after five years. However, the antibiotic duration and organism profile were not associated with treatment succes in their study. This study found that patients with recurrent infection and repeat two-stage revision caused by methicillin-resistant CoNS had a worse survivorship (51% vs. 74%) compared to patients with other organisms. In considering these findings, the aggressive use of oral antibiotics with high bioavailability such as liniezolid and potentially long-term, supressive treatment could be a success factor in patients with resistant PJI caused by CoNS, however further comparative study designs are needed.

We found that patients who had infections caused by so-called difficult-to-treat (DTT) organisms had a similar outcome compared to patients with non-DTT organisms, although these types of organisms were significantly more common at the time of the second PJI. This is in contrast to a study [22] that investigated 59 patients with hip or knee PJI and compared 31 patients with DTT organisms to 28 patients with less resistant organisms. They found that the infection resolution rate was significantly lower in patients with a PJI caused by a DTT organism (69% vs. 88%). However, while they performed a two-stage revision for all patients included, the definition of what qualifies as a DTT organism varies across studies, and their study also considered some *Enterococcus* spp., *Cutibacterium* spp. and some rare organisms such as *Granulicatella* as DTT, while other definitions only consider rifampicin-resistant *Staphylococci,* ciprofloxacin-resistant Gram-negative organisms, and fungal species to be difficult to treat. Furthermore, they included patients with first-time two-stage exchange and did not focus on repeat two-stage revisions. Therefore, it appears plausible that at the time of recurrent chronic PJI, antibiotic susceptibly, when defined as DTT, may not play an overly important role, but other factors superpose it. However, as DTT infection as well as repeat two-stage exchanges are relatively rare conditions, future prospective studies appear warranted.

This study found that Gram-negative organisms are significantly more common at the time of the second PJI, and these patients had a trend toward having worse infection control compared to patients that had Gram-positive or other infections (42% vs. 78%). This relatively poor outcome and potential of Gram-negative pathogens in patients with recurrent infection has also been observed by Karczewski et al. [23]. They investigated 30 patients with hip PJI caused by a Gram-negative organism and found an infection-free implant survivorship of 61% after five years. Furthermore, they noted that 23% had pathogen persistence at further surgeries. While they included 18 two-stage revisions and 12 other revision procedures for Gram-negative PJI, they did not focus on repeat two-stage infections. They reported that 20% of Gram-negative organisms were considered difficult to treat as they were resistant to fluoroquinolones, which is comparable to this study. Although potentially difficult to administer considering this resistance, the authors discuss prolonged suppression therapy as an approach to address the high risk of reinfection and pathogen persistence, which must be emphasized as well, considering our results.

## 4. Materials and Methods

### 4.1. Study Design

The approval of the local ethics committee (2019-042-F-s Ethikkommission der Aerztekammer Westfalen-Lippe und der Westfaelischen-Wilhelms Universitaet Muenster) was obtained before the initiation of this retrospective cohort study. Patients were included if they met the following criteria: history of a completed two-stage exchange arthroplasty for chronic hip or knee PJI at our institution, diagnosis of further periprosthetic joint infection of the same joint analog to the criteria published by the Musculoskeletal Infection Society (MSIS) from 2011 [24], treatment with repeat resection arthroplasty and planned delayed re-implantation at our institution between 2010 and 2017, and a minimum follow-up period of one year [17]. However, patients who did undergo revision surgery or died prior to that were also included. Patients with a prior resection of a bone tumor and subsequent infection were excluded from this study. Using our prospectively maintained institutional joint registry, we identified 281 patients who had undergone a two-stage exchange arthroplasty of a hip or knee prosthesis due to chronic PJI at our institution between 2011 and 2017. Of these, 46 were treated with repeat resection arthroplasty and planned delayed re-implantation between 2011 and 2017 due to chronic reinfection. Furthermore, we excluded 4 patients who had a second two-stage exchange for PJI because they remained culture-negative at the second episode of PJI and at potential repeat revision surgeries. The median follow-up period was 34 months (interquartile range (IQR) 25–45).

### 4.2. Definitions

Success of the repeated two-stage exchange arthroplasty was defined following the Delphi-based consensus definition that includes healed wounds, no further surgical procedure for infection, and no PJI-related mortality [25]. PJI was considered chronic if there were clinical symptoms for more than six weeks or a fistula had formed.

### 4.3. Microbiology Methods

Tissue samples were sent to the hospital’s microbiology laboratory in a sterile plastic container without additives. The samples were homogenized with a scalpel and were cultured on Columbia blood agar, chocolate agar, and Schaedler agar at 37° Celsius for 7 to 14 days depending on growth. Species identification was performed with MALDI-TOF mass spectrometry (Bruker, Bremen, Germany). Difficult-to-treat organisms were defined as *Staphylococci* with resistance to rifampicin, Gram-negative organisms with resistance to fluorochinolones, and fungal organisms.

Data regarding the patients’ surgical history, clinical course, medication, and pre-existing comorbidities were collected from electronic files. An age-adjusted Charlson Comorbidity Index (CCI) was calculated for each patient. Infections were classified as persistent rather than new infections if at least one pathogen that was cultured at the explantation stage of the preceding two-stage exchange arthroplasty was cultured again at the subsequent two-stage exchange or any further revision for PJI failure.

### 4.4. Patient Demographics

The patient demographics of the study cohort at the time of the repeat two-stage exchange are presented in Table 2 and Table 3.

### 4.5. Antibiotic Treatment

All patients underwent a minimum of six weeks of systemic antibiotics based on resistance testing (Table 4 and Table 5). Usually, the first two weeks of systemic antibiotics were administered intravenously in an in-patient setting, followed by four weeks of oral antibiotics in an outpatient setting. After second-stage reimplantation, the same antibiotics were continued until cultures came back negative. If there was organism growth at the time of reimplantation, it was individually discussed if it most likely was a contaminant (a single positive low-virulence organism) or an infection (multiple positive cultures, highly virulent organisms) with the microbiologist. Furthermore, it was discussed with the patient if prolonged antibiotics should be administered in their individual situation. There was no difference in the choice of antibiotic substance or class in between the first and second two-stage revisions (*p* = 0.158).

All patients received local antibiotic treatment by means of a polymethylmethacrylate (PMMA) bone cement spacer (Copal G+C, Heraeus Medical, Wehrheim, Germany), which was handmade using 6 mm titanium rods. The cement contained 1 g of clindamycin and 1 g of gentamicin. For Gram-positive infections, 2 g of vancomycin was added, while 2 g of meropenem was added for Gram-negative organisms. For fungal infections, 600 mg voriconazole was added per 40 g of PMMA. We did not use other forms of local antibiotics such as soaking during the study period.

Frequencies were analyzed and are given as the absolute number and percentages. All numerical variables were tested for normality using the Kolmogorov–Smirnov test. Considering the non-parametric distribution of the tested variables, the median and 25–75% interquartile range (IQR) are presented, and comparative testing was conducted using the Mann–Whitney U-test and McNemar test. Categorical variables were compared using cross-tables and the chi-square test. All values are given with their corresponding 95% confidence intervals (CIs). All *p* values were two-sided, and the *p* value was set at 0.05.

## 5. Conclusions

Changes in organism profile and antibiotic resistance are an important factor in managing patients with recurrent PJI who are planned to undergo repeat stage revision surgery. The outcome can be relatively poor with high risk of reinfection. We recommend considering Gram-negative and polymicrobial PJI at the time of a second two-stage exchange and plan local and systemic treatment accordingly in order to empirically cover these pathogens.

## Figures and Tables

**Table 1 antibiotics-13-00771-t001:** Microbiology findings at first and second two-stage revision.

	First Two-Stage% (n/46)	Second Two-Stage (n/46)
Coagulase-negative *Staphylococcus*	48 (22)	39 (18)
*Staphylococcus aureus*	22 (10)	15 (7)
*Enterococcus* spp.	4 (2)	9 (4)
*Streptococcus* spp.	9 (4)	13 (6)
Gram-negative	11 (5)	28 (13)
*Candida*	4 (2)	8 (4)
others	11 (5) *	20 (9) **

* Corynebacterium, Bacillus, Finegoldia, Dermabacter, Cutibacterium. ** Corynebacterium, Bacillus, Cutibacterium, Bacillus.

**Table 2 antibiotics-13-00771-t002:** Patient demographics.

Variable	% (n/46)
TKA	61 (28)
men	54 (25))
obesity	63 (29)
diabetics	37 (17)
smokers	17 (8)
alcoholics	4 (2)

**Table 3 antibiotics-13-00771-t003:** Patient demographics (metric variables).

Variable	Median (IQR)
body mass index	31 (28–35)
Charlson comorbidity index	4 (3–5)
age at second two-stage revision	74 (65–78)
number of previous revisions	5 (4–7)

**Table 4 antibiotics-13-00771-t004:** Details on systemic intravenous antibiotic treatment.

Antibiotic Substance	First Two-Stage Revision % (n/n)	Second Two-Stage Revision % (n/n)
vancomycin	48 (22)	44 (21)
flucloxacillin	17 (8)	7 (3)
daptomycin	4 (2)	20 (9)
aminopenicillin	6 (3)	14 (6)
cephalosporine	24 (11)	-
meropenem	-	14 (6)

**Table 5 antibiotics-13-00771-t005:** Details on systemic oral antibiotic treatment, combination treatment with mostly rifampicin.

Antibiotic Substance	First Two-Stage Revision % (n/n)	Second Two-Stage Revision % (n/n)
linezolid	44 (20)	35 (16)
clindamycin	17 (8)	-
rifampicin	22 (10)	22 (10)
fluorochinolone	11 (5)	30 (14)
aminopenicillin	17 (8)	24 (11)
fluconazole	-	6 (3)

## Data Availability

Anonymized data sets are available upon reasonable request.

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
