# Peer review of "Do Organism Profile and Resistance Patterns Change between First and Subsequent Two-Stage Revision for Periprosthetic Joint Infection?"

_antibiotics, 2024, doi:10.3390/antibiotics13080771_

Round 1

Reviewer 1 Report

Comments and Suggestions for Authors

The manuscript is well written and organzed. I recommend acceptance in its present form

Author Response

Thank you for the appreciative comment. We have however made several changes based on the other reviewers' recommendations.

Thank you for your work and kind regards

Reviewer 2 Report

Comments and Suggestions for Authors

The paper "Do organism profile and reistance pattterns change between ..." may be suggest interesting result, but in my opinion it is almost impossible to establish the difference in bacterial resistance between the first and the second two-stage revision without knowing what kind/class of antibiotics were used in the first and two-stage revision.

I assume that antibiotics were used during the surgery and also after, but but nowhere is it described, neither the type nor the class. 
It is not described if temporary prostheses soaked in antibiotics were used before inserting the second prosthesis, a method widely used to avoid infection of the second prosthesis.

In essence, to make a description of the difference in antibitoics resistance of bacteria between a first and a subsequent two-stage review is necessary to know which antibiotics these microorganisms were resistant to, and in this paper this important notice is not reported!

      Comments on the Quality of English Language

No comment onn the quality of English langauge.

Author Response

ear reviewer, thank you for your important comments and aspects that you pointed out. We are glad that we are given a chance to adress those in a revision in order to make our manuscript a valuable contribution to the literature

The paper "Do organism profile and reistance pattterns change between ..." may be suggest interesting result, but in my opinion it is almost impossible to establish the difference in bacterial resistance between the first and the second two-stage revision without knowing what kind/class of antibiotics were used in the first and two-stage revision.

I assume that antibiotics were used during the surgery and also after, but but nowhere is it described, neither the type nor the class. 

Thank you for this important comment. We agree with you that details on antibiotic treatment are needed. We added a paragraph as well as a further table to the methods and analyzed whether the chosen substances varied between the two time point (did not significantly). (lines 295ff, Table 4+5)

It is not described if temporary prostheses soaked in antibiotics were used before inserting the second prosthesis, a method widely used to avoid infection of the second prosthesis.

Thank you for this comment, all patient had local antibiotics in form of a PMMA spacer with high dose local antibiotic which was added to the methods. We did not use antibiotic soak or similar local antibiotics during the study period.

In essence, to make a description of the difference in antibitoics resistance of bacteria between a first and a subsequent two-stage review is necessary to know which antibiotics these microorganisms were resistant to, and in this paper this important notice is not reported!

Thanky you, see above, we have added extensively on systemic and local antibiotics.

Reviewer 3 Report

Comments and Suggestions for Authors

Dear Authors of manuscript entitled “Do organism profile and resistance patterns change between first and subsequent two-stage revision for periprosthetic joint infection?” below please find my suggestions and comments regarding with your paper

- Please check the title of your paper.

- Please delete the subsection "Backgroud, Methods, etc...    from the abstract so that you can write one structured paragraph without separation.

- The Introduction section needs to be more developed. Add more recent references! the same think for the Disussion section! no references from the current year 2024 were found in  your manuscript! 

- The Results section: check the title position of the table 1! (above and not bellow ec table) even the forms of the table should be revised.

- The same remark for table 2 and 3

- In Table 1: all the bacterial species should be written in Italic form and the genus should be capitalized! For example: staphylococcus aureus should be: Staphylococcus aureus. Please check this pint in all the manuscript.

- Materials and Methods section: add subsections to this part! it is weakly presented

Author Response

Dear reviewer, thank you very much for working on our manuscript. We are more than glad and grateful to incorporate your valuable suggestions into the manuscript and hope that it therefore becomes a valuable contribution to the existing literature on this rare but very serious issue.

Please check the title of your paper.

Thank you, but we are unsure how the title should be changed.

  • Please delete the subsection "Backgroud, Methods, etc...    from the abstract so that you can write one structured paragraph without separation.

Thanks that has been changed as proposed.

  • The Introduction section needs to be more developed. Add more recent references! the same think for the Disussion section! no references from the current year 2024 were found in  your manuscript! 

Thank you very much, we have updated the references and added papers from 2024 to both sections.

- The Results section: check the title position of the table 1! (above and not bellow ec table) even the forms of the table should be revised.

The same remark for table 2 and 3

Thank you, this has been changed.

  • In Table 1: all the bacterial species should be written in Italic form and the genus should be capitalized! For example: staphylococcus aureus should be: Staphylococcus aureus. Please check this pint in all the manuscript.

Thanks, this has been changed.

Materials and Methods section: add subsections to this part! it is weakly presented

Thank you, this has been changed. Furthermore, we expanded this part by details on antimicrobial treatment.

Thank you and kind regards

Reviewer 4 Report

Comments and Suggestions for Authors

Thank you for this helpful contribution. The authors present a work to answer the question: Do organism profile and resistance patterns change between first and subsequent two-stage revision for periprosthetic joint infection? Antibiotic resistance is an issue in the  treatment of periprosthetic joint infection, and and  the microorganism profile plays a role with potentially more resistant or polymicrobial infections. This is a descriptive study of 46 patients from a single healthcare center with reinfection. The authors concluded that the patients who undergo a repeat two-stage revision for recurrent periprosthetic joint infection have more  polymicrobial and resistant organisms.

In addition to the below comments, the manuscript would benefit from additional proof reading for grammar. I write some comments below that could benefit the article: The author should review the table 1 presentation format. review table presentation format according to journal mdpi.  Tables 2 and 3 can be unified and it is recommended that the subtitle of said table be better explained. Please, review the tables. References: The authors should review journal references (ref 2, 9, 10, 11, 12, ..). They must be written following publication rules.

The introduction presents the topic of study adequately and refers to previous studies, and recent studies 2021. The objective is well formulated and is appropriate to the research.

The analysis of the results is basic descriptive but details relevant aspects of the research. I only have doubts about the small number of the study sample. Do you think the results would be different if the sample size were larger? What if a multicenter study had been proposed? Have they not detailed characteristics of the center where the study was carried out? Although it is true that other previous studies have been developed with a similar number of participants.

It is recommended that you use subtitles in the material and methods section to facilitate the reader's understanding.

Thank you for this invitation to provide a review to evaluate this article for publication. I am delighted to have been invited to review this work.

Author Response

Dear reviewer, thank you for your appreciative review and the important comments on how the manuscript can be approved.

We hope that the changes made, improve the manuscript to qualify for publication.

In addition to the below comments, the manuscript would benefit from additional proof reading for grammar.

Thank you, the manuscript has been proof-read by an native speaker and edited.

:The author should review the table 1 presentation format. review table presentation format according to journal mdpi.  Tables 2 and 3 can be unified and it is recommended that the subtitle of said table be better explained. Please, review the tables.

Thank you, we used the journal template for the tables and added tables regarding antibiotic therapy. We moved the heading to the top as suggested. We chose to keep to tables on demographics as one present numerical data with medians/means and the other frequencies. This is difficult to combine and would be poolry readable in our opionion.

References: The authors should review journal references (ref 2, 9, 10, 11, 12, ..). They must be written following publication rules.

Thank you, we used the template and relied on the MDPI style with endnote. This has been updated and converted.

The introduction presents the topic of study adequately and refers to previous studies, and recent studies 2021. The objective is well formulated and is appropriate to the research.

Thank you.

The analysis of the results is basic descriptive but details relevant aspects of the research. I only have doubts about the small number of the study sample. Do you think the results would be different if the sample size were larger? What if a multicenter study had been proposed? Have they not detailed characteristics of the center where the study was carried out? Although it is true that other previous studies have been developed with a similar number of participants.

Thank you, we absolutely agree with your assessment that there is a need for future larger mutlicenter studies. We have amended the limitations section proposing this.

It is recommended that you use subtitles in the material and methods section to facilitate the reader's understanding.

Thanks, this has been changed.

Thank you again for your fair review of our manuscript

Round 2

Reviewer 2 Report

Comments and Suggestions for Authors

The paper "Do organism profile and resistence patterns change between first and subsequent two-stage revision for periprosthetic joint infections?"results better structured and interesting after revision. The paper brings interesting and important changes especially in the materials and methods section in comparison to the first version. 

Obviously the Authors report that future, multi-institutional studies are necessary to confirm tjìhese preliminaryy data.

Reviewer 3 Report

Comments and Suggestions for Authors

accept

Reviewer 4 Report

Comments and Suggestions for Authors

The authors have made the proposed changes and responded satisfactorily to the comments and questions. The paper is accepted for publication.